# Footprint of sustained poleward warm water flow within East Antarctic submarine canyons

Federica Donda [1] ✉, Michele Rebesco [1], Vedrana Kovacevic [1], Alessandro Silvano [2], Manuel Bensi [1], Laura De Santis [1], Yair Rosenthal [3], Fiorenza Torricella[1], Luca Baradello[1], Davide Gei[1], Amy Leventer [4], Alix Post [5], German Leitchenkov [6,7], Taryn Noble[8], Fabrizio Zgur[1], Andrea Cova[1], Philip O'Brien [9] & Roberto Romeo[1]

The intrusion of relatively warm water onto the continental shelf is widely recognized as a threat to Antarctic ice shelves and glaciers grounded below sea level, as enhanced ocean heat increases their basal melt. While the circulation of warm water has been documented on the East Antarctic continental shelf, the modes of warm water transport from the deep ocean onto the shelf are still uncertain. This makes predicting the future responses of major East Antarctic marine-grounded glaciers, such as Totten and Ninnis glaciers, particularly challenging. Here, we outline the key role of submarine canyons to convey southward flowing currents that transport warm Circumpolar Deep Water toward the East Antarctic shelf break, thus facilitating warm water intrusion on the continental shelf. Sediment drifts on the eastern flank of the canyons provide evidence for sustained southward-directed flows. These morpho-sedimentary features thus highlight areas potentially prone to enhanced ocean heat transport toward the continental shelf, with repercussions for past, present, and future glacial melting and consequent sea level rise.

The East Antarctic Ice Sheet (EAIS) is receiving increasing attention as it represents the largest potential contributor to both past and future sea level rise, containing at present ca. 52 m of Sea Level Equivalent (SLE[1,2]). In some sectors, where the EAIS ice is grounded well below the sea level, rapid ice shelf basal melting is occurring due to intrusions of relatively warm Circumpolar Deep Water (CDW[3–5]), the largest source of heat to the ice sheet[6]. Defining both the extent and the long-term persistence of intrusion of warm CDW on the continental shelf (e.g. refs. 7–9) is critical because it can substantially increase the vulnerability of marine-based sectors of the EAIS, with major consequences

for predicting their future responses to climate warming. Here we focus on the Totten and Ninnis glaciers, which represent key outlets of the two main marine-based sectors of East Antarctica: the Aurora-Sabrina and Wilkes subglacial basins, respectively (Fig. 1).

The Totten Glacier (Sabrina Coast; Fig. 1) has a large marine-based catchment and exhibits among the highest basal melt rates (10.5 m/yr) in the whole East Antarctica[10]. These rates are comparable with those observed in rapidly retreating and thinning sectors of the West Antarctic Ice Sheet (WAIS[11–13]). Totten Glacier alone drains a volume of ice above flotation equivalent to >3.5 m of sea-level rise[1,14], similar to the

[1]National Institute of Oceanography and Applied Geophysics—OGS, Borgo Grotta Gigante 42/c, 34010 Sgonico, Trieste, Italy. [2]School of Ocean and Earth Science, University of Southampton, University Road, Southampton SO17 1BJ, UK. [3]Department of Marine and Coastal Sciences, Rutgers, State University of New Jersey, New Brunswick, NJ 08901, USA. [4]Geology Department, Colgate University, Hamilton, NY 13346, USA. [5]Geoscience Australia, GPO Box 378, Canberra, ACT 2601, Australia. [6]The All-Russia Scientific Research Institute for Geology and Mineral Resources of the Ocean, St. Petersburg, Russia. [7]Institute of Earth Sciences, St. Petersburg State University, 199034 St. Petersburg, Russia. [8]Institute for Marine and Antarctic Studies, University of Tasmania, Hobart, TAS 7001, Australia. [9]Earth and Environmental Sciences, Macquarie University, Sydney, NSW 2109, Australia. ✉e-mail: fdonda@ogs.it

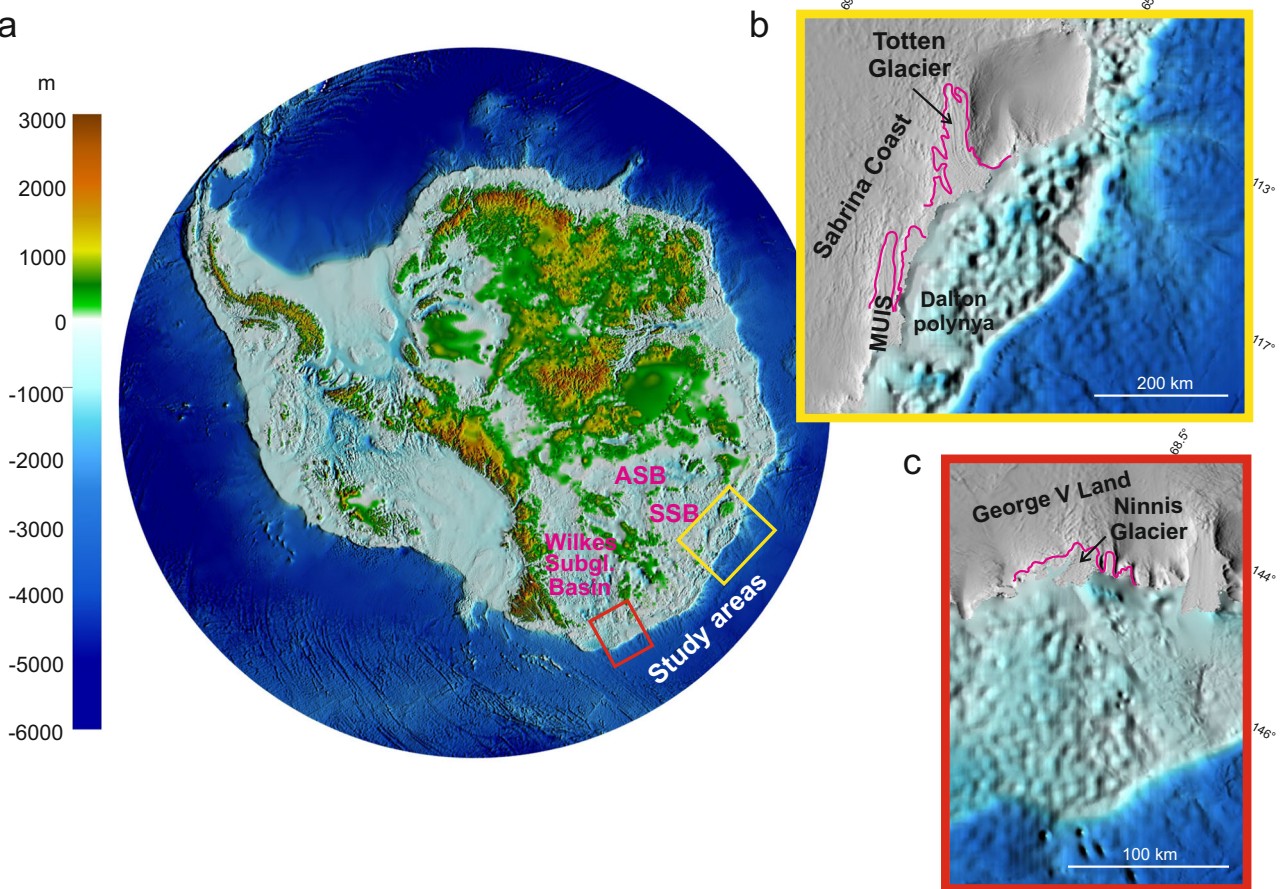

**Fig. 1 | Location of the study areas. a** Subglacial bed elevation derived from Bedmap 2 grids[71]. ASB: Aurora Subglacial Basin (after ref. [72]); SSB: Sabrina Subglacial Basin (after[73]) (**b**). Subglacial bed elevation of the Sabrina Coast marked by the yellow frame on the overview map. **c** Subglacial bed elevation at the Ninnis

Glacier location marked by the red frame on the overview map. MUIS: Moscow University Ice Shelf. Terrestrial terrain map from the Reference Elevation Model of Antarctica (REMA[74]). Magenta lines: grounding lines at the Totten Glacier and MUIS (after Li et al. [15]) and at the Ninnis Glacier (after ref. [27]).

---

entire WAIS[10]. The rapid basal melting, thinning, and acceleration of the Totten Glacier, particularly evident since 1969[15], are likely associated with ocean heat transported by CDW intrusions onto the continental shelf[16–21], with the morpho-bathymetry playing a key role in steering warm waters from the shelf break toward the glacier[14,19–22]. On the Sabrina continental shelf, CDW accesses the ice-shelf cavity at the glacier front[3]. The thick layer of CDW (-1000 m) is transported efficiently towards the Sabrina Coast shelf break by southward currents on the rim of a quasi-stationary cyclonic eddy with a spatial scale of 100–200 km, guided by the underlying bathymetry[20]. Over the last three decades of observations, there is no evidence for the production of Dense Shelf Water on polynyas found on the Sabrina Coast continental shelf (Fig. 1)[3,4,16,17,21], contrary to polynyas from other Antarctic regions. Thus, CDW intrusions, allowing warm water to reach the ice shelf cavities, can cause rapid basal ice melting[4], potentially leading to the acceleration of the glacier and reduced buttressing[18]. Totten Glacier grounding line retreat is predicted to increase 3.5 x by 2100 in response to changes in the inflow of warm water onto the Sabrina continental shelf[23,24].

Ninnis Glacier (George V Land) holds 95 cm Sea Level Equivalent[10], and buttresses the ice sheet draining part of the Wilkes Subglacial Basin (WSB[25]), the largest marine-based drainage basin in East Antarctica. During the last century, from 1913 to 1962, the Ninnis Glacier tongue retreated about 90 km[26], possibly as a result of periodic calving events[27]. Ninnis Glacier occupies a landward sloping basin[1], is grounded below sea level[28], and holds potential for rapid retreat should the CDW access the ice sheet margin[10,29]. Relatively warm water has been

found near the shelf break[30], and the presence of basins up to 1400 m deep on the continental shelf could convey warm water toward the ice shelves[28,31,32]. Similar to the Aurora-Sabrina subglacial basins, the Wilkes Subglacial Basin is thus vulnerable to the intrusion of warm CDW across the continental shelf, possibly causing irreversible retreat[33]. The Ninnis Glacier is currently in contact with cold dense shelf water[34,35], forming Adélie Land Bottom Water (25% of the global volume of Antarctic Bottom Water[36], and references therein). However, changes in atmospheric and ocean circulation may have driven retreat in the past, as indicated by numerical ice sheet modeling ([37] and references therein) and by ice and marine sediment cores (refs. [38–42]).

Here we document submarine canyons on the East Antarctic continental rise that host sediment drifts formed by predominantly southwards currents. This evidence argues for the existence of a persistent quasi-barotropic (i.e. from surface to bottom) southward flow, which also favors the transport of the CDW layer toward the continental shelf. This, in turn, may have influenced Totten and Ninnis glaciers in the past, and increases their vulnerability to future ocean warming.

## Results

Sub-bottom CHIRP (Compressed High Intensity Radar Pulse) data collected offshore the Sabrina Coast reveal the occurrence of sedimentary deposits on the eastern flanks of the Maadjit and Manang canyons, from ca. 119°E to ca. 121°E (Fig. 1), in water depths ranging from 3250 to 3550 m (Fig. 2). These mounded deposits, ca. 2000–3500 m wide and ca. 50-80 m thick, are asymmetric, with a

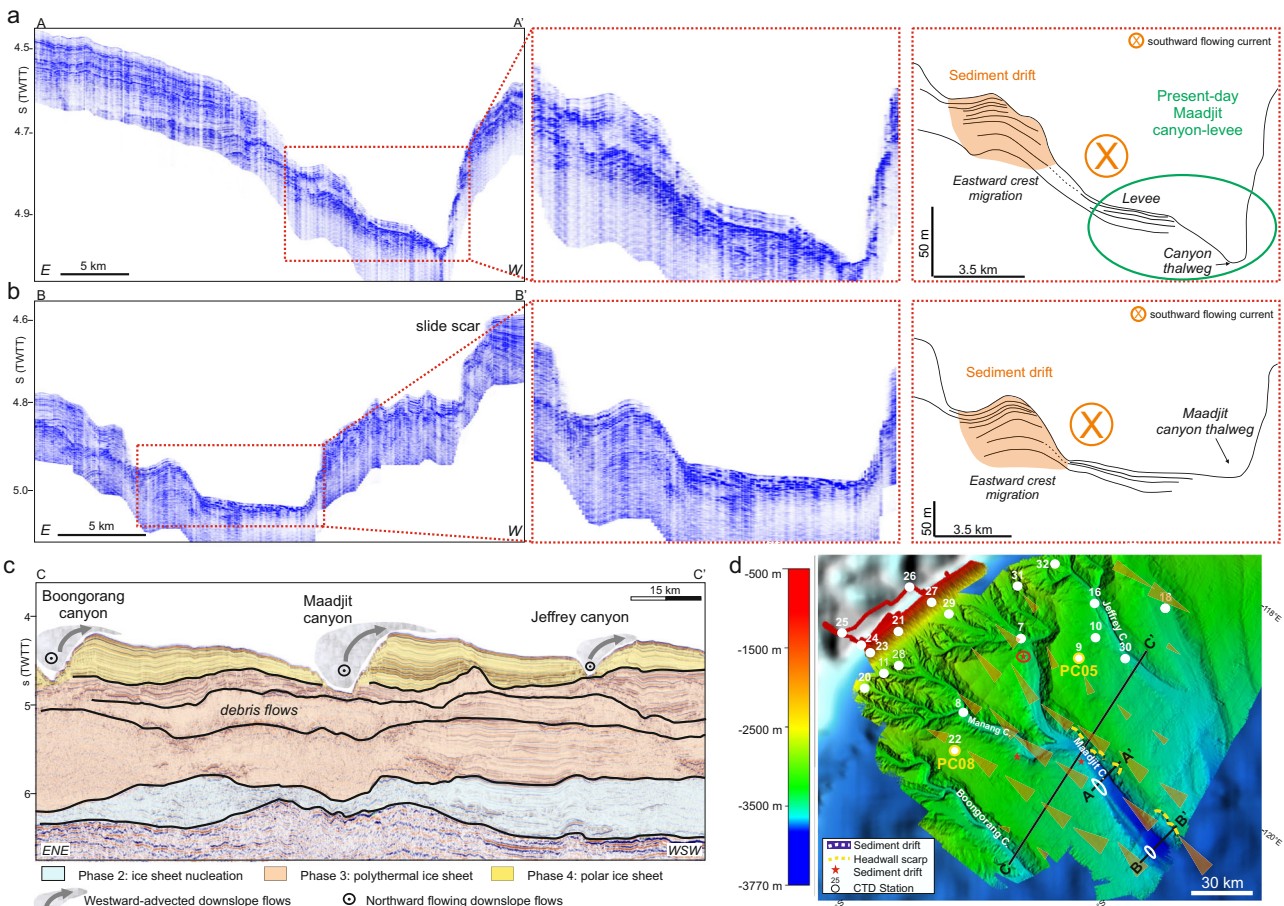

**Fig. 2 | CHIRP (Compressed High Intensity Radar Pulse) profiles collected off Totten Glacier. a** Part of the sub-bottom CHIRP profile 110 collected on the Sabrina Coast continental rise and seismostratigraphic interpretation highlighting the relationship between the present day Maadjit canyon-levee system, represented by the elongated feature visible on the morpho-bathymetry map (**d**) and the sediment drift deposited on its eastern flank. The sediment drift appears to be coeval or even younger than the levee deposit, and shows an eastward crest migration; (**b**) part of the sub-bottom CHIRP profile 72 collected on the Sabrina Coast continental rise and seismostratigraphic interpretation. The sediment drift is thicker than to the south (80 m vs 50 m; the time-depth conversion is shown in the "Methods" section). In this area, the present-day Maadjit canyon is lacking the related eastern levee, possibly since downslope flows progressively lose competence as they reach deeper areas. To the west, abrupt reflection truncations suggest the occurrence of a slide scar (see also **d**). **c** Multichannel seismic profile RAE5108 showing the overall "paleo configuration" of the Sabrina Coast continental rise. Here sediment-laden downslope gravity currents were deviated to the west by the combined effect of the slope currents and the Coriolis force when a highly dynamic ice sheet, reaching the continental margin, led to turbidite flows across the continental slope (modified from[43,51]). **d** Location map of the sub-bottom CHIRP and multichannel seismic profiles; white dots indicate the position of Conductivity-Temperature-Depth (CTD) vertical profiles gathered in January–February 2017; the location of sediment cores PC05 and PC08 coincides with CTD09 and CTD022 stations, respectively; the encircled red star shows the location of the sediment drift shown in the Supplementary Information; light orange arrows represent time-mean (2011–2018) surface velocity vectors from ref. 20. The location of the slide scar recognizable on the CHIRP profile shown in (**b**) is also marked. The white ellipses outline the inferred shape of the sediment drifts shown in (**a, b**).

steeper flank facing the thalweg of the canyon and a gentler eastern flank, and show a slight eastward crest migration. They appear to grow above a higher amplitude mounded surface, possibly representing the harder substratum of a terrace similar to those described by O'Brien et al.[43]. Internally, their thinly stratified acoustic facies includes low to medium amplitude, laterally continuous seismic reflections, subparallel on the gentler side, while converging before terminating on the steeper side (Fig. 2). The configuration of the seismic reflections and the overall morphology of these sediment bodies are diagnostic of bottom current-related deposits[44–46] and are similar to those of small scale sediment drifts observed in high- and low latitude continental margins[47]. Similarly to the Sabrina Coast, a sediment drift has been found on the continental rise in front of the Ninnis Glacier at around 148°E. Morpho-bathymetry data reveal that this area is cut by a canyon system resembling the Maadjit and Manang canyons, with dendritic-like branches in the slope that converge into a single canyon. The thinly stratified, asymmetric sediment drift lies on the canyon's eastern flank

at a depth of 3050 m, is ca. 2750 m wide and ca. 40 m thick, and shows a slight eastward crest migration (Fig. 3).

Based on their position, progressive crest migration and shape, we argue that these sediment drifts formed because of southward flowing bottom currents focused on the eastern side of the canyon, potentially under the influence of the Coriolis force (e.g. ref. 48). The Coriolis effect is known to strongly influence the internal flow structure of oceanic currents, especially at high latitudes, deflecting them towards the left in the southern hemisphere. As a consequence, sediment-laden currents flowing from the shelf break towards the rise would cause formation of sedimentary deposits on the left, i.e., the western flank of the canyons[49]. This is inferred to have occurred off the Sabrina Coast during glacial maxima and especially during the build-up of canyon-levees in the so-called "Phase 3" of ref. 50 and references therein (Fig. 2c), tentatively dated early-to-late Miocene in age, when a highly dynamic, polythermal ice sheet delivered large amounts of sediments on the continental slope and rise via slumps and turbidity

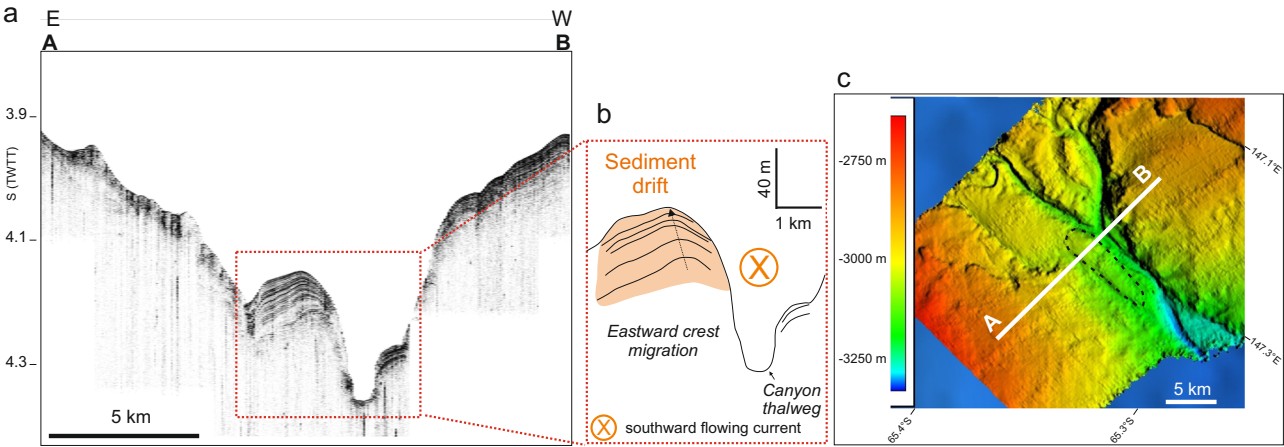

**Fig. 3 | TOPAS (Topographic Parametric Sonar) profiles collected off Ninnis Glacier. a** Part of a TOPAS profile collected on the continental rise in front of the Ninnis Glacier in the frame of the PNRA (National Antarctic Research Program) Collapse cruise. **b** Seismostratigraphic interpretation highlighting the internal configuration of a ca. 40 m thick sediment drift (the time-depth conversion is shown in the "Methods" section) on the eastern flank of the canyon; in this area, poorly constrained by oceanographic measurements, this feature represents a morpho-sedimentary proxy for southward-directed bottom flows. **c** Location of the sub-bottom TOPAS profile; the dotted ellipse outlines the inferred shape of the sediment drift shown in (**a**, **b**).

flows ([51] and references therein). The highly asymmetric canyon-levees formed in the continental rise since the mid-Cenozoic reflects non-uniform deposition from downslope gravity flow deviated to the west by the combined effect of the Antarctic slope currents and the Coriolis effect[43,51]. These processes and related canyon-levee configuration constitute the so-called "paleo configuration". Downslope processes still contributed to shaping the distal margin architecture even during and after the transition to polar ice sheet regimes (i.e., "Phase 4" of ref. [50] and references therein; Fig. 2c), by transferring sediments from the shelf to the rise area through the long-lived canyon systems ([51]; Fig. 2). Conversely, the sediment drifts that we discovered and discuss here, imaged on the high-resolution CHIRP and TOPAS profiles, are explained with a different mechanism. In the "present day configuration", the main canyon is smaller compared to the "paleo configuration" and is entrenched to the west. The weaker turbidity currents possibly generated throughout the Phase 4 were not able to develop on the steep, up to 700 m-high, "paleo" western levee, being thus only capable to build a small levee to the east (Fig. 2a and Supplementary Fig. 1). On the Sabrina Coast continental rise, the sediment drifts build-up further to the east of this small eastern levee, and their internal reflection configurations indicate that they are coeval or even younger than the levee deposits (Fig. 2a and Supplementary Fig. 1).

Off the Sabrina Coast, ocean currents have been estimated using in situ and satellite measurements[20,52]. These observations show a southward flow in the Maadjit canyon from the surface to the sea floor (with velocities up to 10 cm/s in the bottom 100 m of the water column). Such southward flows are associated with a deep-reaching semi-permanent cyclonic eddy[20,50,53]. It is recognized as playing a key role in southward transport of offshore CDW, which is then able to access the continental shelf at around 120°E ([18–21]). Vertical profiles obtained from conductivity-temperature-depth (CTD) casts carried out during the IN2017-V01 cruise (see Fig. 2d for their location) highlight the presence of CDW, with potential temperature > 0 °C, salinity > 34.5 (potential density > 27.7 kg m$^{-3}$), and dissolved oxygen < 240 uM, between 300 m and 1200 m depth over the continental slope and rise. At the head of the Maadjit and Manang canyons, CDW is found at stations 28 and 29 and at stations 7 and 8. Similarly, a signal of warm water is found in the 100 m-thick bottom layer on the continental shelf at stations 25 and 26 directly to the south of the canyons head (Fig. 4). A signature of Antarctic Bottom Water (AABW), with potential temperature < 0 °C, salinity > 34.63, dissolved oxygen > 240 uM, neutral density > 28.27 kg m$^{-3}$ is instead clear over the slope at stations 28 and 29 below 1300 m, and

at stations 7 and 8, below 1800 m depth. Approaching the shelf break (i.e. the head of the canyons), the CTD profiles coherently show that the water column between 1000 and 2500 m is progressively more influenced by colder, fresher and more oxygenated waters of shelf-slope origin, modifying the properties of CDW accessing the continental shelf (Fig. 4b–d).

## Discussion

Here we propose that two key factors regulate the asymmetric sedimentation within the canyons in the study areas: (1) The bathymetric constraint imposed by the canyons support preferential, almost depth-independent (barotropic) southward flow[20]; (2) The southward flow tends to be focused on the eastern side of the canyons, allowing the growth of sediment drifts on the eastern flank of the canyons (Fig. 5). In other areas of the Antarctic margin, i.e. in the Ross Sea, southward currents impinging the continental shelf are associated with the overspill of dense waters from the shelf[54]. This water masses exchange is not occurring off Sabrina Coast and it has not been observed since three decades of measurements, although it may have occurred in the past.

Along the Sabrina Coast, the Coriolis effect is important in shaping the overall morphology of the continental slope and rise ([43,51] and references therein). Northward flowing turbidity currents displaced by the Coriolis effect gave rise to giant asymmetric sediment ridges during the "paleo configuration"[43,51,55]. The entrenchment of the "present day" canyon against the western " paleo" levee (Fig. 5), and the seismostratigraphic characteristics of the canyon thalweg, being constituted by a high amplitude seismic facies with no sub-bottom reflections (Figs. 2 and 3), suggest that the canyon floor contains coarse sediments, deposited by "bedload dominated" flows[56]. The canyon progressively migrates towards west, as commonly observed at high southern latitudes (see fig. 11 f of ref. [56]). Off the Sabrina Coast, the emplacement of debris flows has probably deleted the record of the progressive canyon axis migration (Fig. 2c), which is recognizable in the other canyon systems (see Fig. 2a of ref. [57]). The turbidity currents, less vigorous than those formed under the"paleo configuration" and northward flowing within the narrower canyon thalweg, led to the formation of the most recent, smaller levee (Fig. 2a, b). The shape, internal configuration, position and progressive crest migration of the sediment drift to the east of the canyon levee indicate that an opposite, southward flowing bottom current shape these deposits. At this time, there are no sediment archives of the eastern drift deposits in the study

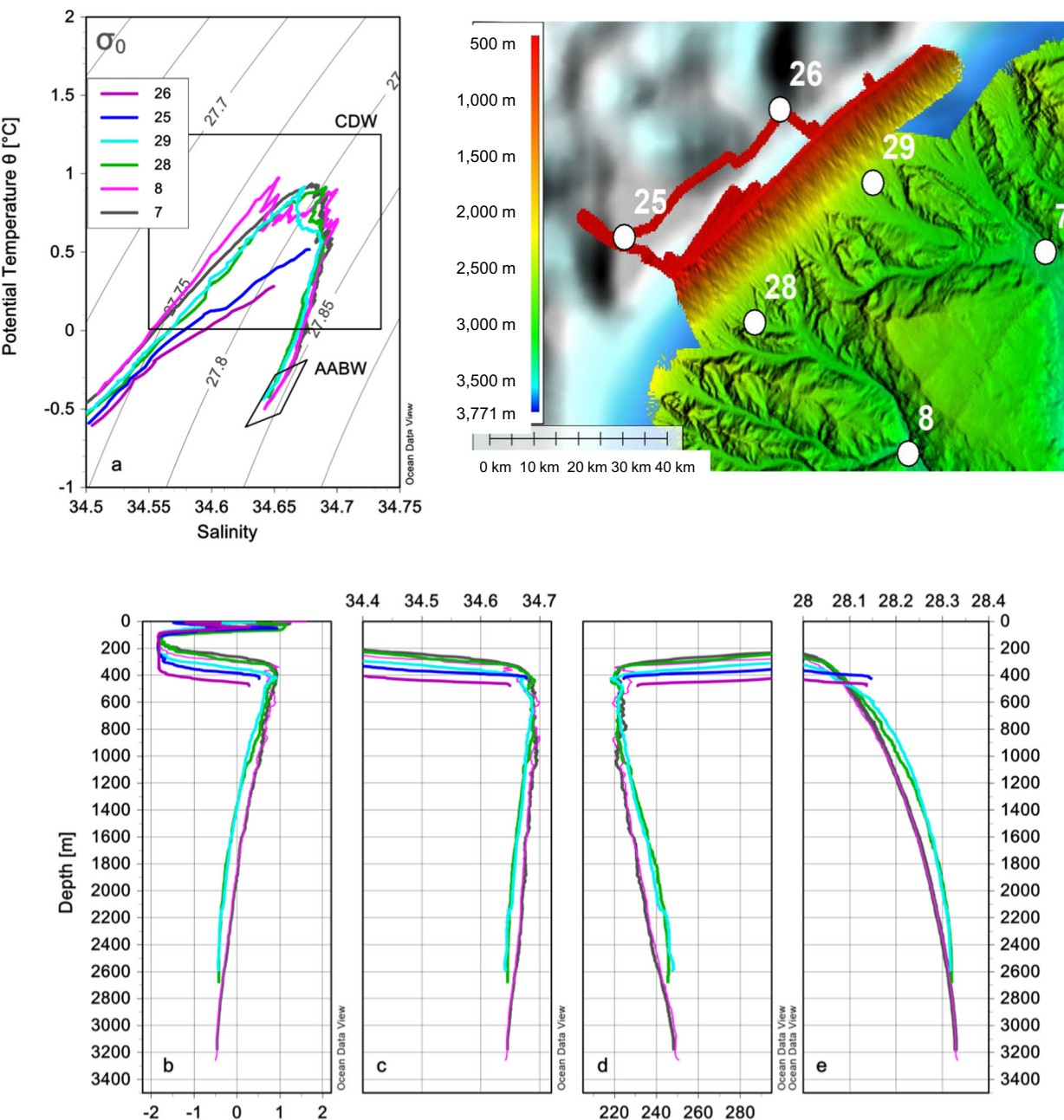

**Fig. 4 | Sea water properties at selected CTD stations with emphasis into the intermediate and deep layers (>200 m depth). a** potential temperature Θ (°C)-salinity scatter plot with isolines of potential density anomaly $\sigma_O$ (kg/m³) referred to 0 dbar (for more details see ref. 21). Vertical profiles of (**b**). potential temperature Θ, (**c**) salinity, (**d**). dissolved oxygen concentration (μM), and (**e**). neutral density $\gamma^n$. The $\gamma^n$ values > 28.27 kg/m³ are typical of the AABW. The position of CTD casts is shown in Fig. 2d (note that casts 25 and 26 are situated on the continental shelf).

regions, with only one kasten core on the Sabrina margin (IN2017_C020_KC08; 3354 m) located at the base of the western edge of the Maadjit canyon, reflecting sedimentation regimes alternating between turbidite deposition and biogenic accumulation of diatom mats[43]. While the sediment drifts have not been directly sampled, two piston cores collected in the area provide indications of sedimentation rates. One core collected on the sediment ridge to the west of the Maadjit Canyon (Core IN2017 C012_PC05; 64°40.517'S, 119°18.072' E; 3099 m; Fig. 2d), revealed 100 Kyr glacial-interglacial cyclicity over the past 350 kyr, with average sedimentation rates of 4.6 cm/kyr for the last 300 kyr[58]. The second core collected to the east of the sediment ridge located between Manang and Boongorang canyons (Core IN2017_C025_PC08, 64° 57.0'S, 120° 51.6'E; 2787 m; Fig. 2d), shows

sedimentation rates of 3.3–5.7 cm/kyr for the late deglaciation and Holocene[59]. Comparable sedimentation rates have been found in deep-sea Antarctic sediment drifts, including offshore the Antarctic Peninsula, where sedimentation rates of 5–12 cm/kyr were recorded on the crest of drift 7[60]. Sediment rates of 5–10 cm/kyr have been reported from sediment cores at 3000 m water depth, from the western flank of the WEGA channel in the offshore of the George V Land[61]. In a different, shallower setting, sedimentation rates of ca. 14 cm/kyr are estimated for the plastered drift on the continental shelf of the Amundsen Sea Embayment[62]. This drift is formed by southward inflow of CDW, i.e. by a process similar to what we suggest for the formation of the sediment drifts that we discovered in a deeper setting within the canyons. Taking into account that the sediment drifts we identified in our study areas

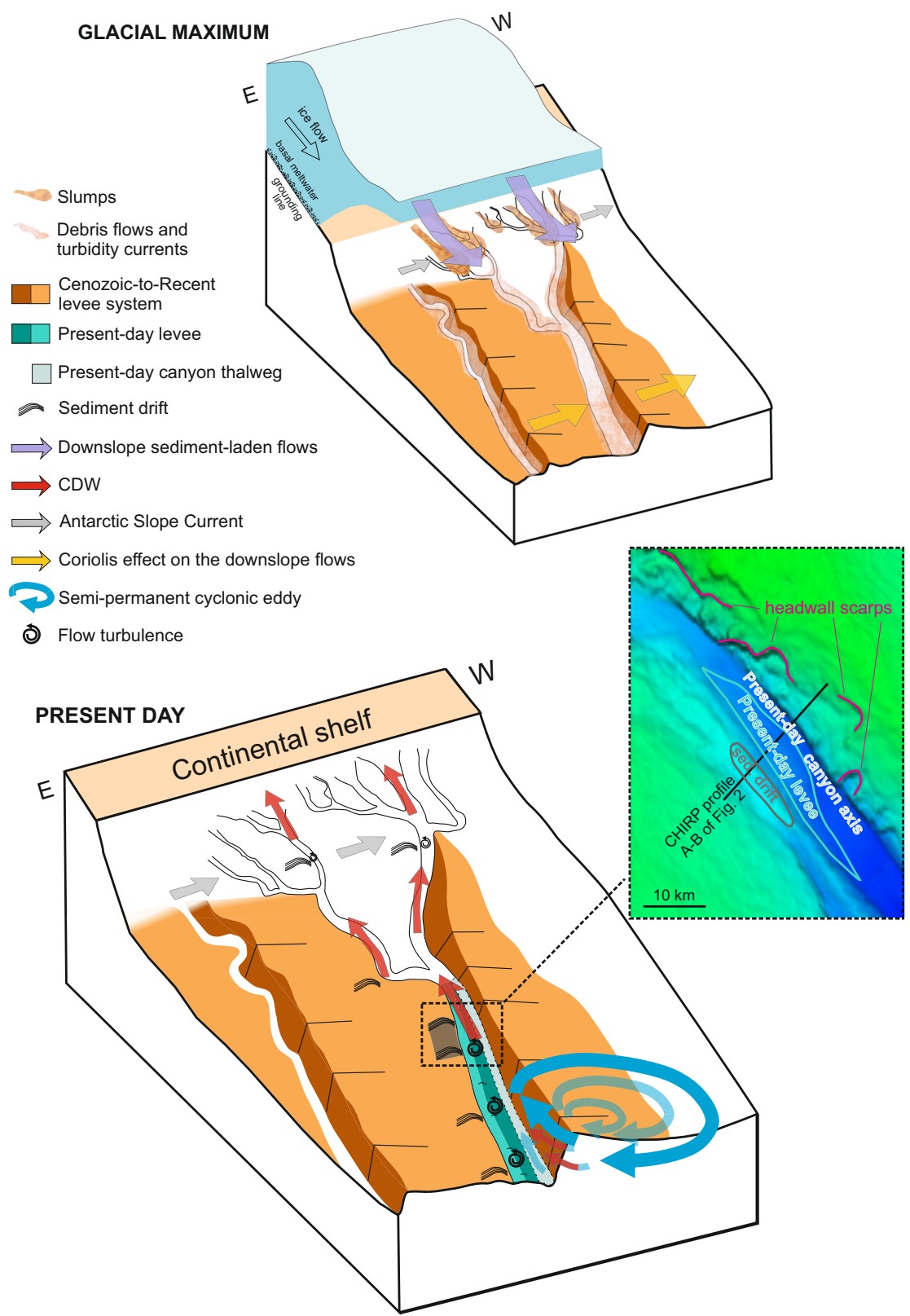

are 40 (to 80) m thick, and assuming a sedimentation rate of a few cm/kyr, i.e. 3.3 cm/kyr like the minimum recorded by piston core PC08[59], the drifts likely formed in the last million of years, and thus contain a record of sustained southward-flowing currents, conveying the warm CDW toward the continental shelf.

Southward-flowing currents off Sabrina Coast are associated with the southward component of the deep-reaching quasi stationary cyclonic eddies that have a spatial scale from 100 to 200 km[20]. A further contribution could be related to the southward (i.e. leftward) deflection of the westward Antarctic Slope Current once it encounters a canyon, similar to what is observed in major bathymetric troughs around Antarctica[34]. Once formed, the southward-flowing currents are strongly influenced by bathymetric constraints[20,21]. In nearby Vincennes Bay, CDW also intrudes across the continental shelf through a

**Fig. 5 | Cartoon showing the two configurations of the Sabrina Coast depositional environment: the paleo configuration during glacial periods and the present day configuration.** During glacial maxima, when the ice sheet grounded at the continental shelf, enhanced turbidite flows occurred and were predominant across the continental slope and rise, where they were deviated to the west by the Coriolis force, leading to the formation of the large-scale asymmetric sediment ridges in between the main canyons[51,57]. Currently, downslope flows are strongly reduced or even absent, with the canyons funneling the southward flowing currents derived by the semi-permanent, deep-reaching cyclonic eddies and, possibly by the Antarctic Slope Current (ASC), carrying the warm Circumpolar Deep Water (CDW) to the shelf break[20]. The occurrence of dozen meters-thick sediment drifts on the canyon's eastern flank indicates sustained southward-directed bottom flows. Given the observed barotropic circulation, the sediment drifts represent a footprint for transport of warm CDW across the continental rise and slope and ultimately its intrusion onto the continental shelf.

small canyon-like structure on the slope[63]. At the Lützow-Holm Bay, southward-flowing warm waters are transported by the cyclonic Weddell Gyre toward the Shirase Glacier Tongue leading to high basal melt rates[5,64]. Such warm waters are conveyed by a trough that extends to the continental slope[5], where it is possibly associated with a canyon system (see Fig. 1a of ref. 5). We thus maintain that canyons represent preferential pathways for enhanced CDW intrusion onto the continental shelf. When entering the deeply incised canyons, vigorous flows (i.e. with the measured 10 cm/s velocity[52]) become more turbulent, potentially creating small-scale eddies that could impact the rugged seafloor (Fig. 5; [43]). In Antarctica, it has already been shown that topography plays a major role in sediment drift formation, indeed[65]. On the Sabrina Coast continental rise, the first baroclinic Rossby radius has values of about 5 km (based on vertical gradients of ocean density from CTD data, see Fig. 4, and corresponding Brunt-Väisälä frequencies). For the deep canyons (stations 7 and 8) such value is comparable or smaller than the width of the Maadjit canyon, which is 5.3–9.5 km wide, at the upper and lower rise respectively[43]. The complex interactions of internal oceanographic processes and bottom morphology provide a physical mechanism for the formation of the sediment drifts, with a key role played by the southward flow associated with the semi-permanent large-scale eddies. Small-scale eddies can affect the deposition of the fine grained suspended sediment, shaping the seafloor by creating and maintaining sediment drifts over time[66].

Several areas of the Antarctic margin are still largely unexplored, and in some of them only one type of data, e.g., seismic data has been collected. This is the case for large parts of East Antarctica, where seismic data and "only" a few oceanographic measurements and sediment cores are available. In these areas, the occurrence of sedimentary features similar to those we unveiled here, could represent a key morpho-sedimentary proxy documenting sustained southward bottom flows and associated transport of warm CDW across the continental rise and slope and ultimately its intrusion onto the continental shelf.

Understanding the pathways and intensity of currents flowing toward the continental shelf is key for evaluating both the past and the future vulnerability of marine-grounded glaciers. Sediment drifts have been found also on the continental shelf of the Amundsen Sea Embayment (West Antarctica), where they have been interpreted as plastered sediment drifts formed as a consequence of sustained southward inflow of warm CDW[62]. On the George V Land continental shelf (East Antarctica), the Mertz Drift formation appears to be strongly linked to the CDW inflow within the George V Basin[67,68]. The canyons and sediment drifts investigated in this work indicate that long-lasting flow of CDW on the continental slope and rise occurred offshore of both the Aurora-Sabrina, and Wilkes subglacial basins. Our results highlight that these basins are vulnerable to ocean-driven basal melting, consistent with paleo-marine sedimentary records revealing up to 700 km of ice sheet retreat at the Ninnis Glacier caused by warmer Southern Ocean temperatures within the past 350 kyr[41]. This work demonstrates that the Antarctic canyons serve as "effective" conduits, capable of funneling warm CDW towards the continental shelf, and are key regions for investigating the mechanisms governing CDW intrusions and the role it plays in Antarctic ice sheet (in)stability with implications for Global Mean Sea Level.

## Methods
### SABRINA COAST
Sub-bottom profiler CHIRP, morpho-bathymetry, and CTD data were collected during January–March 2017 onboard the research vessel RV Investigator. Details of the survey are available at the Australian Marine National Facility web site under the survey number IN2017-V01 (https://www.cmar.csiro.au/data/trawler/survey_details.cfm?survey= IN2017_V01[69]).

Sub-bottom profiler CHIRP data were acquired with a Kongsberg SPB120; a linear chirp with a sweep of 2.5–6.5 kHz was used. The pulse length was 6 (in the study area) to 12 milliseconds (during transits in deep water). A gain of 6 dB was applied to the data, followed by a gain correction, matched filter, instant amplitude processing, and a time variable gain to enhance sub-bottom reflections. The dataset was recorded and presented as two-way time sections, saved as an envelope in segy format (Society of Exploration Geophysics Y format) and Kongsberg proprietary.raw files (O'Brien et al.[43]). Segy files were then uploaded into a HIS Markit Kingdom software project and viewed in Seisee. The envelope data has a low frequency and contains only positive values, without phase information[70].

Key seismic profiles were then re-processed, to enhance the internal configurations of the identified sedimentary features. The envelope raw data were processed using Seismic Unix (https://wiki.seismic-unix.org/start). Due to the great depth (most of the sediment drifts lie at ca. 3500–3650 m depth), the original profiles have a large intertrace (20–22 m). Therefore, the signal was spatially reduced by creating an additional track as a mixture of the neighboring traces. In areas with steep slopes to avoid aliasing effects, the traces were rectified in time before mixing and then shifted.

The gain (compensation of transmission loss, absorption and spherical divergence) was calculated and applied by inverting the amplitude decay curve. A water column muting was performed before creating the envelope data for interpretation.

To obtain an estimation of the thickness of the sediment drifts, we applied a time-depth conversion, by using the seismic velocities derived from each high-resolution multichannel seismic profile collected in the study area during the IN2017_V01 cruise, as described in 53. Seismic velocities from the seafloor to 100 ms depth range from 1571 to 1603 m/s.

The morpho-bathymetry was acquired using two multibeam echosounders, a Kongsberg EM122 and an EM710. The former is a 12 kHz full ocean depth multibeam echosounder and was operated at all depth ranges on the Sabrina Seafloor Survey. The EM710 is a high resolution 70–100KHz sounder used for mapping on the continental shelf and upper slope to depths of up to 2000 m. Sound velocity corrections were undertaken using data from CTDs and eXpendable BathyThermographs (XBTs). Data have been processed within Caris HIPS and SIPS version 9.1.9. Further details about both data collection and processing are provided in 54.

The oceanographic data presented in this study are part of the 31 conductivity–temperature–depth (CTD) vertical profiles (labeled 1–11, 13–23, 25–33) acquired in the study region (Fig. 2) during the IN2017_V01 cruise using a Seabird SBE 9plus (pump-controlled) probe and SBE 11plus V2 Deck Unit. The probe was integrated with an additional sensor for measuring dissolved oxygen concentration (DO, SBE43), and an altimeter (PA500). A SBE32 Carousel Water Sampler

was equipped with 36 Niskin bottles (OceanTest Equipment Inc. Florida; 12 L capacity each one, mounted on the rosette frame). Further details about both data collection and processing are provided in 21.

## GEORGE V LAND

About 4000 km of sub-bottom Topas Kongsberg PS18 0.5-6 kHz, ca. 10.000 km² of multibeam Kongsberg MBES EM304 Frequency range (26–34 KHz), were collected during the XXXVIII Italian Expedition on board of R/V Laura Bassi in the framework of PNRA Collapse Project (P.I. L. De Santis), in water depth ranging from 700 – 3500 meters. The investigated area lies between ca. 65°S and 64°S and 145°E and 152°E (https://www.pnra.aq/index.php/it/project/528/cook-glacier-ocean-system-sea-level-and-antarctic-past-stability).

The main acquisition parameters of the sub-bottom TOPAS profiles are briefly reported here. Transmitter: Normal transmit mode; external trigger mode; CHIRP pulse form; start frequency 2 kHZ; stop frequency 6 kHz; CHIRP length 20 ms; output decibel −9°; output level 12.6%; auto beam control; transducer speed 1500 m/s. Receiver: manual delay control; sample rate 64 kHz; trace length 500; HP-filter 2.0 kHz.

Sub-bottom profiler data were acquired using a keel mounted, parametric Kongsberg - Geoacoustic TOPAS echosounder, connected to the Motion Reference Unit of the ship to automatically compensate for the pitch, roll, and heave components; in order to maximize the penetration, a 20 ms long chirp signal, with an upsweep ranging from 2 kHz to 6 kHz, was used. The data, recorded in two way time and double polarity, were saved in the proprietary Kongsberg raw format; they were then converted to SEGY through the TOPAS PS18 software. The data were processed with the SCHLUMBERGER Vista 19 software. The first step was the application of the static corrections (stored in an appropriate header) for a proper time delay compensation; a gain function and a spiking deconvolution (with an operator length of 2 ms) were successively applied to compensate for the amplitude decay and to increase the temporal resolution, respectively. Finally, a top (sea bottom) mute followed by the application of a Hilbert envelope function was adopted.

To obtain an estimation of the thickness of the sediment drifts, we applied a time-depth conversion, by using the seismic velocities derived from each high-resolution multichannel seismic profile collected off Sabrina Coast.

Morpho-bathymetry data (MBES) were acquired between 06/03/2022 and 18/03/2022 by means of the 30 kHz Hull mounted Kongsberg EM304 1°(tx) x 2°(rx) in mainly uncharted areas and processed by means the SW 'QPS Qimera'. Kongsberg EM304 MBES system uses separate transmit and receive transducers in a Mills Cross configuration. The transmit fan generated is divided into individual sectors for unique control, enabling active stabilization in real time to correct for any yaw and pitch movement of the vessel, while roll stabilization is applied on the receiving beams. Up to 800 individual beams are normally available, or up to 1600 if the dual swath mode is enabled. In dual-swath mode two individual transmitting fans are generated with a small difference in tilt to provide a constant sounding separation along track, resulting in a dense sounding pattern on the seafloor. R/V Laura Bassi acquired MBES data using two different software packages at the same time, i.e. both with 'QPS Qinsy', the main navigation program used during the campaign and with 'SIS5', supplied by Kongsberg, the same brand that supplied the 2 MBESs equipped with the ship, the Kongsberg EM304 and EM2040c. As well as Qinsy data format, the KMall KONGSBERG logging format supplied by 'SIS5' contains already calculated latitude, longitude, depth, time, and ellipsoidal height, which allowed us to combine the two above mentioned data formats within the used 'QPS Qimera' processing program. Velocity profiles used during the acquisition worked quite well, as did the calibration performed before the start of the acquisition. The dataset was initially examined line by line and cleaned up by manual despiking using the

SWATH EDITOR, SLICE EDITOR and 3D EDITOR to remove most of the residual outliers not filtered out by the software during the acquisition phase, and then, in a second phase, the same lines were re-examined and further cleaned up using some statistical data validation functions provided by "QPS Qimera" (CUBE, SPLINE). Following the analysis of the preliminary DTMs obtained, lines that could potentially cause some residual noise were again inspected and surgically treated. Finally, 2 different Digital Terrain Models were created with grids of different sizes (20 m and 100 m) depending on the depth investigated. Part of the morpho-bathymetry data shown here were re-gridded by using a grid cell 50 m wide.

## Data availability

Sub-bottom profiler CHIRP, CTD, and morpho-bathymetry data collected off Sabrina Coast are available at https://www.cmar.csiro.au/data/trawler/survey_details.cfm?survey=IN2017_V01          Sub-bottom TOPAS profiles and morpho-bathymetry data collected off George V Land are under embargo and will be made available within 4 years of collection. For information please contact Dr. Laura De Santis, the PI of the PNRA COLLAPSE project, in the frame of which the data have been collected. The multichannel seismic line RAE5108 is available through the Antarctic Seismic Data Library System (SDLS) at https://sdls.ogs.trieste.it/cache/index.jsp.

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

## Acknowledgements

We thank Dr. F. Ferraccioli for the comments to the first version of the manuscript. We thank the technical and scientific staff of the two cruises IN2017-V01 and PNRA COLLAPSE onboard the research vessels Investigator and Laura Bassi, respectively. Part of this work is supported by the Australian Government's Australian Antarctic Science Grant Program (AAS #4333) and by the Australian Government through the Australian Research Council (DP170100557) and the PNRA national Italian project PNRA19_00022 "Cook glacier-Ocean system, sea LeveL and Antarctic Past Stability" (COLLAPSE). We acknowledge HIS Markit Global Sàrl for providing an academic license for Kingdom-Seismic and Geological Interpretation Software, and the Antarctic Seismic Data Library System (SDLS) for providing open access to the multichannel seismic data.

## Author contributions

F.D., M.R., V.K., A.S., M.B.: conceptualization, methodology, data analysis, writing—original draft, writing—review and editing, visualization; D.S.L.: data collection, writing—original draft, writing—review and editing; Y.R.: conceptualization, writing—original draft; F.T.: writing—original draft; L.B.: data processing; D.G.: data processing; A.L.: writing—original draft, writing—review and editing; A.P.: data collection and curation, writing—original draft; G.L.: data collection; writing—original draft; T.N.: writing—original draft; F.Z.: data collection and processing; A.C.: data collection and processing; visualization; P.O'B.: writing—original draft; data collection; data curation, funding acquisition; R.R.: data collection and processing; visualization.

## Competing interests

The authors declare no competing interests.
