## [Peer Review File · Nature Communications]

Footprint of sustained poleward warm water flow within East Antarctic submarine canyonsREVIEWER COMMENTS

Reviewer #1 (Remarks to the Author):

The manuscript submitted by Donda et al. documents the presence of small drift-like features on the east side of canyons in the Sabrina Coast region of East Antarctica. The authors infer that these have been formed by 'sustained' southward (up-slope) flowing warm water masses associated with warm Circumpolar Deep Water, and that these show long-term heat transport towards the East Antarctic ice shelf.

The paper addresses a very important topic. Clearly, understanding the history of warm-water incursion towards the ice shelf around Antarctica is important to understand potential current and future changes that could impact melt rates. The authors propose an interesting hypothesis, with recent oceanographic data showing the presence of semi-permanent eddies in the region (Hirano et al., 2021), and with analogues elsewhere in Antarctica, such as the Ross Shelf (Morisson et al., 2020), showing focused poleward warm-water transport along canyon systems.

However, I have several concerns about the manuscript as it is presented. The major issues are as follows:

Interpretation of sediment drifts in the canyons: "The configuration of the seismic reflectors and the overall morphology of these sediment bodies are diagnostic of bottom current-related deposits" (lines 94-95) – This is an overstatement, based on the information presented. First, the CHIRP profile is relatively low quality, and the interpretation in Fig 2.b is ambiguous. Could this be a slump feature within the canyon for example? Or some aggradation on a terrace within the canyon? The text infers that there are more than one of these, but only one is shown in each canyon. This feature is also not clear on the MBES data, and I would expect that the authors show a much higher-resolution, annotated version of this data to justify their explanation. The single colour image does not show these 'drifts' at all clearly. This is crucial to the argument of the manuscript, and the authors should provide a much more convincing case that these are indeed sediment drifts, and document their lateral extent and locations more clearly. As a more general point, the only original data presented by the authors are two CHIRP profiles, and MBES data at a very small scale in the Maadjit canyon region – not sufficient to justify the interpretation presented.

Inference of southward flowing bottom currents: "The sediment drifts that we identify on the eastern levee of the present-day canyon-levee systems can only have been shaped by a southward flowing bottom current." (Lines 112-113) This again seems to me to be an oversimplification and an overstatement. If these are indeed drifts, they form within the canyons. As the canyons are several hundred metres deep, any downslope currents (mainly turbidity currents) would have to have flow thicknesses higher than the canyon depth to experience flow stripping and sedimentation on the outer levees, which in this case should be on the west of the channels due to Coriolis. However, if the flows are confined to the channel, this would not occur. In this case, the hydrodynamics of the flow would be more complicated, and could result in net deposition on the terrace to the east, without the need for Coriolis-driven transport to the left, up-channel. The authors do refer to the analogue data from the Ross Sea (Morisson et al., 2020), which is intriguing, but they don't show any morphological evidence of bedform migration, for example, that would justify that these drifts are forming directly from upslope-flowing currents.

In addition to this, the argument would benefit from more observational data of up-canyon flow from ADCP data (for example; data from Hirano et al., 2011 are for surface flow). I note that the 'Yamazaki et al., 2023' paper that is cited for ADCP data does not appear in the reference list so this data is not available for context. The reference to Hirado et al., for intermediate water depth velocities are actually for surface flow, and an assumption that near-barotropic flow would result in broadly similar velocities at greater depth, but they don't have any actual velocity measurement for the deepwater

water masses.

Driver for southward flowing bottom currents: the main analogue that the authors use to justify their interpretation for strong southward flowing currents is from Morisson et al., 2020, where the up-canyon currents form specifically in response to cascading dense shelf water. But the authors state that "In contrast with other Antarctic regions with polynyas, there is no evidence over the last three decades of Dense Shelf Water (DSW) production in the Dalton Polynya (lines 59-61). This appears to be a problem for their process interpretation, as the currents associated with the eddies documented by Hirado et al. do not appear to be strong enough to entrain sediment and form large contourite drifts. However, numerical modelling results in the Morisson et al., paper do indicate strong DSW production in the study area. This needs to be addressed in any future revisions.

Age of drifts: the authors state that "The seismostratigraphy reveals that sediment drifts are coeval or even younger than the levee deposits on the western flanks" (line 97-98), referring to previously published interpreted seismic profiles. However, the terraces/canyon drifts are not easily resolved on the multichannel seismic profiles. The CHIRP profiles in Figs 2 and 3 are of higher resolution, but relatively low quality, and the drifts are separated from the levees by canyon walls dominated by erosion. It is very difficult to correlate the stratigraphy between the drifts and levees, or to tell which is older, and this statement is unjustified based on the evidence presented. Thus the "fossil configuration" and present-day configuration model in Fig. 5 seems to be speculative. Furthermore, the age model presented is difficult to justify. The authors refer to a piston core a long distance to the west of the study area, but there are other cores much closer to the study location – for example, the PC-05 core published in O'Brien et al., 2020, which is dated. Other cores in or near the canyons should also be considered – sediment accumulation rates in the canyon and on a drift system are likely to be substantially different to those on the open slope. The authors should take care to justify the age model, and acknowledge uncertainties.

Other minor issues include:

- References and values quoted not always appropriate or correct. For example, they state that "Totten Glacier alone drains a 49 volume of ice above flotation equivalent to more than 3.5 m of sea-level rise (Rignot et al., 2019), similar to the 50 entire WAIS (Greenbaum et al., 2015)." However, the value of 3.5 m for Totten is from Greenbaum et al (Rignot et al., quote a value of 3.85 m). The Greenbaum paper focuses specifically on the Totten Glacier, so is inappropriate as a citation for the WAIS.
- Figure 1, the bathymetry is not very clear – consider using colour spectrum, as well as shaded relief. Consider a cross section showing shelf and subglacial topography, grounding lines, etc.
- Figure 2. Line location symbology are different and hard to see.
- There are a number of spelling and grammatical issues that should be addressed.

Reviewer #2 (Remarks to the Author):

Footprint of sustained poleward warm water flow within East Antarctic submarine canyons

The data this manuscript have a relative high quality taking into account the remote area and the difficult to obtained such type of data. These data were obtained with the appropriate techniques as ultra-high resolution seismic (Sub-bottom CHIRP). The main fortress of the manuscript is the potential significance of the results. In this way, authors point out to periodic intrusion of relatively warm deep water onto the continental shelf as a potential cause of melting of ice-caps in Antarctica. This introduce that fact the deep oceanic circulation of warm currents around the Antarctica as a threat to

Antarctic ice shelves ice-caps and glaciers grounded below sea level.

The major ice-sheets around Antarctica (e.g. Weddell & Ross Sea Ice-Caps) acts as the main source of cold deep waters distributed around the world's oceans, playing an important role in "cooling" the Earth during the last geological history, Miocene to present. After my own experience of having carried out eight oceanographic expeditions to Antarctic, I support that role of deep currents in ice-cap melting as observed in the Weddell Sea and, even more, into their role in the distribution of Antarctic deep waters as forming the great system of oceanic's world circulation.

The intrusion of warm water onto the continental, as proposed in this manuscript by accelerating the melt of ice-caps might increase the cold deep water as the formation of Antarctic Bottom Waters (AABW) and their influence in the global ocean circulation. At geological scale, during deglacial Pleistocene times, the rapid retreat of the Antarctic ice-caps have been explained by increasing of temperatures in the southern oceans, and finally resulting rapid sea-level rise as consequence of the rapid retreat of massive ice-caps.

The only weakness that I observe in this manuscript is how to explain the water depth as much of 3,050 meters (4.2 s Two-way Travel Time) of the formation of the plastered drifts into deep incised S to N canyons as consequence of along-slope currents as the Circumpolar Deep Water (CDW). The plastered drift seems clearly associated to "upwelling currents" rising along a S to N submarine canyon and deflected to left by Coriolis effect. Authors explains these upward currents as results of semi-permanent cyclonic gyres formed by the interaction of cross-slope CDW current with the incision of the deep submarine canyons.

In my opinion, it seems much more difficult to explain the plastered drifts formed along the S-N canyons as results of orthogonal currents flowing in along-slope direction than from "upwelling" of N-S sourced deep waters.

It has been also proposed other mechanism to explain the spatial and temporal connection between upwelling of CDW and cascade of DSW (Dense Shelf Water) transport formed beneath ice-sheet. In this way, as a pulse of DSW descends the continental slope, the dense water displaces less dense water within the canyon allowing upwelling of CDW warm waters (see Morrison et al. 2020).

Probably, authors should improve the explanation(s) and discussion on the mechanism proposed dealing with the formation of cyclonic gyres enough or not to transport "upwelling" CDW warm waters onto the shelf.

Nevertheless, given the potential significance of the results, I strongly recommend the publication of this manuscript after making some further explanation and discussion on the proposed mechanism of intrusion/upwelling of warm waters onto the shelf.

Reviewer #3 (Remarks to the Author):

Review for Nature Communications 463358, "Footprint of sustained poleward warm water flow within East Antarctic submarine canyons". Overall, I see no Major Weakness in the manuscript or research presented therein. First I'll summarized some overall thoughts before several minor weakness and line-by-line comments/edits.

Overall, this work represents an incremental but important step in Antarctic ice-ocean interactions, i.e. presenting new evidence for sustained meridional flow in two East Antarctic systems. The material is mostly presented in a clear fashion, and the research methods are sound - I do not have any criticisms of either their stratigraphic or hydrographical interpretations. In fact, I think they are

exceptionally clear examples of the hypothesis processes at play, and illustrates how to tie oceanography and solid-earth disciplines. However I have a few concerns about the overall impact of the work. As the authors themselves state, their work confirms that southward flow can be inferred using multichannel seismic/CHIRP, something that has been postulated by theory and confirmed elsewhere in Antarctica. While this is certainly a finding worthy to share with the community, for me it does not transform our thinking of the processes at play or the fate of glaciers in East Antarctica. The reader is told that the poleward flow of CDW to the ice sheet is critical for projecting future ice mass loss and sea level rise, but in its present form, the only result from this paper related to these topics are that the identified features "represent at least a ca. 440 to 1300 kyr long record of southward flowing currents". How does this help improve/understand projections of future glacial melt and SLR? Furthermore, although the CTD analysis is fairly sound, single hydrographic profiles cannot in any way 'confirm' a sediment record thousands of years long. In summary, this work, as submitted, provides very clear and compelling arguments for extended periods of southerly flow along these canyons, but I came away unconvinced of its importance to the stated motivations (i.e., ice melt and sea level rise)

Ln 24: Subject?

Ln 25: What is the justification of looking at these two glaciers?

Ln 26: What about the canyons isy being 'documented'?

Ln 31: Does it say something about SLR directly, or just indirectly through glacial melt?

Ln 59-61: This sentence is ambiguous. As written it sounds like there are "Dalton Polynyas" in "other Antarctic regions". Also, are you saying that the Dalton Polynya hasn't formed? Hasn't produced DSW?

Ln 69: The bathymetry resolution in some areas seems too low. Can you use Bedmap3? It appears to be available via SCAR as of this summer.

Also, the 'Depth' scale doesn't mean much, and topography looks 'inverted' in 1b inset, maybe due to hillshade angle?

Finally, perhaps this Figure 1 is not needed? All the reader gets (in its present form) is the location and trough/slope shape overview. Readers already get the bathymetry in more detail in Figs 2 and 3. So consider eliminating Fig 1 and add a circum-Antarctic Overview inset to each Figs 2 and 3?

Ln 74: Really 'one of the largest'?

Ln 93: "are"?

Ln 104: "argue"?

Ln 105: How wide is the canyon/current? Is the Rossby radius of deformation relevant here? I know it is difficult to calculate given the dearth of measurements, but can it be estimated?

Ln 113: "Only"? Is this strong of a statement defensible?

Ln 119: Capitalize? And add "."

Ln 131: In addition to the cartoon in Figure 5, the discussion of these drift and levee features could benefit from a plan view zoom-in of a single feature (like 10 km wide version o)

Ln 143: It would be VER helpful for a zoomed in location plot for the CTDs shown in Figure 4, perhaps

as an inset?

Ln 144: "Potential" temperature? (theta)

Ln 146: Apostrophe?

Ln 153: As written, the "properties" access the continental shelf. I suggest "modifying the properties of CDW accessing the continental shelf."

Ln 163: Is this specific to "East" Antarctica?

Ln 165: "barotropic"?

Ln 173: Remove comma.

Ln 184: Interesting! I was wondering about 'timing' earlier in the paper. Is there a way to elevate this point or move it up?

This leads me to ask, can you compare southward flow during the different epochs? Even if relatively? Was it stronger/weaker or more/less consistent?

Ln 196: What does "rugged" mean here? Have you made the case why it's relevant?

Ln 199: I don't get the point of this short paragraph. Either clarify, or eliminate.

Ln 212: This confuses me, as throughout the rest of the paper we are told that this southward flow is extremely persistent. If so, what is its role in "destabilization"?

Ln 219: In general I really like this cartoon, however some things are not very clear, especially the "sediment drift" features.

Also, "glacials" to "glacial periods"?

Ln 222: Remove "were" and "giant"?

Ln 224: "flowing flows"? Redundant.

Ln 237: This list is hard to follow and therefore has little meaning.

Ln 255: Add space.

Ln 280: This seems like a less-detailed, redundant paragraph. Suggest removing it.

Ln 295: I think "easy" is a bit too colloquial (and subjective) for a research article.

Ln 296: Too qualitative for me. Can you put numbers on this noise? If not, remove.

Ln 303: Define "DTM".

Ln 310: "going to be"? This is not good practice.

REVIEWER COMMENTS

Reviewer #1 (Remarks to the Author):

The manuscript submitted by Donda et al. documents the presence of small drift-like features on the east side of canyons in the Sabrina Coast region of East Antarctica. The authors infer that these have been formed by 'sustained' southward (up-slope) flowing warm water masses associated with warm Circumpolar Deep Water, and that these show long-term heat transport towards the East Antarctic ice shelf.

The paper addresses a very important topic. Clearly, understanding the history of warm-water incursion towards the ice shelf around Antarctica is important to understand potential current and future changes that could impact melt rates. The authors propose an interesting hypothesis, with recent oceanographic data showing the presence of semi-permanent eddies in the region (Hirano et al., 2021), and with analogues elsewhere in Antarctica, such as the Ross Shelf (Morisson et al., 2020), showing focused poleward warm-water transport along canyon systems.

However, I have several concerns about the manuscript as it is presented. The major issues are as follows:

Interpretation of sediment drifts in the canyons: "The configuration of the seismic reflectors and the overall morphology of these sediment bodies are diagnostic of bottom current-related deposits" (lines 94-95) – This is an overstatement, based on the information presented.

Following the Reviewer's comment, we added a new CHIRP profile and expanded both the "Results" and the "Discussion" sections in order to improve the information on the studied features.

First, the CHIRP profile is relatively low quality, and the interpretation in Fig 2.b is ambiguous.

We have re-processed two CHIRP profiles, i.e., the profile 110 already shown in the previous manuscript version, and a new profile 72_004, collected further to the north. See the next response for clarity on how the reprocessed data improve our interpretation.

Could this be a slump feature within the canyon for example? Or some aggradation on a terrace within the canyon?

We thank the Reviewer for this comment, which forced us to perform, for the Sabrina coast dataset, a re-analysis of the CHIRP profiles, particularly focused on the re-processed lines, and of the multibeam bathymetry. Our re-analyses thus complement the outcomes of the previous geomorphological study of O'Brien et al., (2020), where the first geomorphological map of the Sabrina Coast continental slope and rise was discussed. In the study area, those Authors already evidenced how the western flank, i.e. the steeper flank of the Maadjit Canyon is commonly scalloped, and represents areas of sediment mass movement, where arcuate features constitute headwall scarps of slumps; downslope of them, slump deposits are found (O'Brien et al., 2020). The new CHIRP profile we have included in this revised version clearly shows the prominent slide scar on the

western flank of the Maadjit canyon (Fig. 2b), whereas the revised Supplementary Figure 1 outlines the hummocky internal configuration characterizing the slided material (the western flank of the Maadjit canyon), compared to the well stratified features, which we interpret as sediment drift. In fact, on the eastern flank of that canyon, CHIRP profiles and multibeam bathymetry do not reveal the occurrence of such prominent headwall scarps and slump features. The feature we interpret as sediment drifts is well stratified and progressively growing with an eastward crest migration, which is more clearly recognizable after the data re-processing. The elongated feature within the Maadjit canyon, more clearly visible in the new Supplementary Figure 2, possibly represents an ancient terrace, above which sediment drifts locally form. In fact, based on Fig. 6 of O'Brien et al. (2020), terraces have been already identified along the adjacent Jeffrey Canyon (see Figs. 6 and 10a of O'Brien et al., 2020), and are characterized by "low-relief hummocky surfaces" (O'Brien et al., 2023), i.e., they appear very different to the well-stratified features discussed in this study. We have modified the text accordingly.

The text infers that there are more than one of these, but only one is shown in each canyon.

For the area located offshore Totten Glacier, we have inserted a new CHIRP profile, i.e. line 72_004, collected within the same canyon system (i.e., Maadjit Canyon) further to the north. All the three profiles, i.e., those presented in Figs. 2a and 2b and that shown in Supplementary Figure 1 have been reprocessed in order to outline the internal reflectors configurations.

This feature is also not clear on the MBES data, and I would expect that the authors show a much higher-resolution, annotated version of this data to justify their explanation.

In this revised version of the manuscript, we have inserted a revised version of the multibeam bathymetry, with (Fig. 2) and without annotations (Supplementary Figure 2)

The single colour image does not show these 'drifts' at all clearly. This is crucial to the argument of the manuscript, and the authors should provide a much more convincing case that these are indeed sediment drifts, and document their lateral extent and locations more clearly. As a more general point, the only original data presented by the authors are two CHIRP profiles, and MBES data at a very small scale in the Maadjit canyon region – not sufficient to justify the interpretation presented.

We recall here the previous comment: For the area located offshore Totten Glacier, we have inserted a new CHIRP profile, i.e. line 72_004, collected within the same canyon system (i.e., Maadjit Canyon) further to the north. All the three profiles, i.e., those presented in Figs. 2a and 2b and that shown in Supplementary Figure 1 have been reprocessed in order to outline the internal reflectors configurations. MBES data are now shown in Figures 2 and 3 and in the Supplementary Figure S2 with a colour spectrum, which emphasizes the morphobathymetric setting.

Inference of southward flowing bottom currents: "The sediment drifts that we identify on the eastern levee of the present-day canyon-levee systems can only have been shaped by a southward flowing bottom current." (Lines 112-113) This again seems to me to be an oversimplification and an overstatement. If these are indeed drifts, they form within the canyons. As the canyons are several hundred metres deep, any downslope currents (mainly turbidity currents) would have to have flow thicknesses higher than the canyon depth to experience

flow stripping and sedimentation on the outer levees, which in this case should be on the west of the channels due to Coriolis.

We do agree with the Reviewer comment: our statement at lines 112-113 needed to be deeper discussed. This was not possible in the first version of the manuscript, because of the limited number of words allowed in Nature Geoscience, where we submitted the paper in a first instance. Here, we improved this part by adding several statements concerning the relationship between the sediment drifts location and bottom current regime, in the two different configurations, i.e. the “Paleo configuration” (former “fossil configuration”) and the present-day one. Moreover, we agree with the Reviewer comment “As the canyons....due to Coriolis”, and, based on that, that’s exactly what happened when the “Paleo configuration” was shaped: turbidity currents were predominant especially when the ice sheet was able to reach the shelf break. These downslope fluxes were then deviated to the west by both the Coriolis effect and, possibly, the westward-flowing Slope Current, leading to flow stripping and sedimentation on the western levee more than on the eastern one (please see the asymmetry of the levees in the three channel systems shown in Fig. 2 c). The drifts are thus produced by less powerful upslope flows deviated to the east and depositing the sediment suspended within the canyon.

However, if the flows are confined to the channel, this would not occur. In this case, the hydrodynamics of the flow would be more complicated, and could result in net deposition on the terrace to the east, without the need for Coriolis-driven transport to the left, up-channel. The authors do refer to the analogue data from the Ross Sea (Morisson et al., 2020), which is intriguing, but they don’t show any morphological evidence of bedform migration, for example, that would justify that these drifts are forming directly from upslope-flowing currents.

These Reviewer comments highlight that even here we were not clear enough. We do agree with their comment, indeed: we suggest that in the present day configuration, where the channel is small compared to the “Paleo configuration” and entrenched to the west, the downslope flows were (are?) much more weaker than during the shaping of the “Paleo configuration”, and thus they were not able to exceed the western, huge “paleo” levee, being then only capable to build the small levee to the east (please see Fig. 2 a and Fig. Supplementary 1). The features we interpret as sediment drifts lie further to the east of this small levee, and thanks to the data reprocessing, the migration of the sediment drifts crest toward east is now more recognizable. We added some statements in order to clarify all these aspects. Concerning the reference of Morisson et al., 2020, we realized that it does not fit as an analogue because the southward flows in the Ross Sea derive from the gravity currents due to dense water formation on the shelf, which does not occur in our study area. In the Ross Sea, the southward flow on the eastern side of the trough is a response to the northward flow associated with dense water overflow on the western side. Therefore it is a different system.

In addition to this, the argument would benefit from more observational data of up-canyon flow from ADCP data (for example; data from Hirano et al., 2011 are for surface flow). I note that the ‘Yamazaky et al., 2023’ paper that is cited for ADCP data does not appear in the reference list so this data is not available for context. The reference to Hirado et al., for intermediate water depth velocities are actually for surface flow, and an assumption that near-barotropic flow would result in broadly similar velocities at greater depth, but they don’t have any actual velocity measurement for the deepwater water masses.

We agree with the reviewer that the text was not clear enough in terms of available observations and the features of the flow over the slope off the Sabrina Coast. Here a large scale (100-200 km) quasi-stationary eddy creates a persistent southward flow in the canyon region. Hirano et al. 2021 in Figure 3 combine surface velocity derived by satellites with sub-subsurface geostrophic velocity (derived by hydrographic data) to obtain the full-depth profile of absolute velocity (not only the surface component). They show how southward

velocities (up to >0.1 m/s) are observed from the surface to the seafloor (barotropic). This is confirmed by Yamazaki et al. (under review-in Progress in Oceanography) using ADCP data near the seafloor. We have modified the text to better highlight all these points.

Driver for southward flowing bottom currents: the main analogue that the authors use to justify their interpretation for strong southward flowing currents is from Morisson et al., 2020, where the up-canyon currents form specifically in response to cascading dense shelf water. But the authors state that “In contrast with other Antarctic regions with polynyas, there is no evidence over the last three decades of Dense Shelf Water (DSW) production in the Dalton Polynya (lines 59-61). This appears to be a problem for their process interpretation, as the currents associated with the eddies documented by Hirado et al. do not appear to be strong enough to entrain sediment and form large contourite drifts. However, numerical modelling results in the Morisson et al., paper do indicate strong DSW production in the study area. This needs to be addressed in any future revisions.

As mentioned above, the comment of the Reviewer helped us to realize that Morrison et al. (2020) is not a great analogue because in the Ross Sea southern flowing currents are associated to downslope, northward flowing currents related to dense water formation on the shelf. However, Morrison et al (2020) state the following “The DSW is deflected to the western side of the canyons as it descends, due to the Coriolis force, while the CDW ascends in the center or eastern side of the canyons, depending on the narrowness of the particular canyon”. This statement is important for our discussion. Offshore Totten Glacier, no dense water is currently produced, so that downslope flows are expected to be strongly reduced or absent. Similarly to Lützw-Holm Bay, where southward-flowing warm waters are conveyed by the cyclonic Weddell Gyre toward the Shirase Glacier Tongue leading to high basal melt rates (Hirano et al., 2020). Offshore Totten Glacier, warm CDW is transported toward the south by the cyclonic eddy system identified by Hirano et al. (2021). Yamazaki et al. (under review- in Progress in Oceanography) show how southward velocities >0.1 m/s are observed near the seafloor using ADCP data. This flow velocity is similar to that found on the Drift 7 in Antarctic Peninsula (0.06 m/s; Camerlenghi et al., 1997), which is longer and (150 km long, 70 km wide and up to 700 m high) higher than the sediment drifts in our study areas.

Age of drifts: the authors state that “The seismostratigraphy reveals that sediment drifts are coeval or even younger than the levee deposits on the western flanks” (line 97-98), referring to previously published interpreted seismic profiles. However, the terraces/canyon drifts are not easily resolved on the multichannel seismic profiles.

Drifts are not recognizable on the multichannel seismic profiles, due to their narrow width and the low resolution seismic data, which makes it difficult to discern these features on that dataset. Instead, CHIRP and TOPAS profiles collected offshore Totten and Ninnis glaciers respectively, allow identification and characterisation of these previously undiscovered depositional bodies. We decided to insert the already published seismic profile RAE 5108 and related interpretation to help the Reader understand the difference between the “paleo configuration”, characterized by the giant channel-levee systems recognizable on that multichannel seismic line, and the “present day configuration”, where channels and related levees are strongly reduced in size compared to the former.

The CHIRP profiles in Figs 2 and 3 are of higher resolution, but relatively low quality, and the drifts are separated from the levees by canyon walls dominated by erosion. It is very difficult to correlate the stratigraphy between the drifts and levees, or to tell which is older, and this statement is unjustified based on the evidence presented. Thus the “fossil configuration” and present-day configuration model in Fig. 5 seems to be speculative.

In our opinion, the reprocessing of the data allows us to better image the seismic reflector configuration and the relationship between the levee deposits and the sediment drift. Moreover, we have improved the description and discussion about the two main configurations in the results and in the discussion sections, and we think the sketch shown in Fig.5 is more readable now.

Furthermore, the age model presented is difficult to justify. The authors refer to a piston core a long distance to the west of the study area, but there are other cores much closer to the study location – for example, the PC-05 core published in O’Brien et al., 2020, which is dated. Other cores in or near the canyons should also be considered – sediment accumulation rates in the canyon and on a drift system are likely to be substantially different to those on the open slope. The authors should take care to justify the age model, and acknowledge uncertainties.

We agree with the Reviewer comment: we were not clear enough in the mentioning of that sediment core, which is exactly that the Reviewer suggests to consider, i.e., piston core PC05. We added this information in the text. Moreover, a new piston core collected in the study area in the frame of the same survey, i.e., PC08, has been recently published (Sadatski et al., 2023). Similarly to PC05, this core has been collected on the crest of a sediment ridge located between the Maadjit and the Boongorang canyons. The sedimentation rates at the location of PC08 core are 3.3 to 5.7 cm/kyr, similar to those found at the PC05 core. We agree with the Reviewer comment that sediment rates in the canyon and on a drift system are likely to be different. What makes the difference is that the thalweg of a canyon is generally affected by episodic coarse grained bedload (highly erosive in the upper course of the canyon and resulting in high accumulation rates only locally and at the mouth of the canyon), whereas on drifts, levees, and other elevated ridges adjacent to canyons the fine grained suspended load can be deposited in a relatively continuous way with moderate sedimentation rates. Sedimentation rate on the crest of sediment drift 7 (Antarctic Peninsula) is 5 cm/kyr, whereas on the same drift but closer to the Alexander Channel it is of 9-12 cm/kyr (Venuti et al., 2011), thus being (both of them) comparable with those measured in the study area. Also, sedimentation rates estimated for the plastered drift on the continental shelf of the Amundsen Sea Embayment formed by the southward inflow of CDW are ca. 14 cm/kyr (Uenzelmann-Neben et al., 2022). In all these cases from different Antarctic sediment drifts and ridges, the sedimentation rates are of few cm/kyr (from 5 to 14 to be precise) and we don’t have any element for inferring a significantly different sedimentation rate. At the moment, in our knowledge, no analogues of our study cases do exist, thus highlighting the novelty of our contribution, and, at the same time, the need of further investigations, which we are confident to perform in the framework of the new, recently approved project PNRA DIONE (Dynamic behavior of the East Antarctic Ice Sheet in the Sabrina Coast; PI: F. Donda)

Other minor issues include:

- References and values quoted not always appropriate or correct. For example, they state that “Totten Glacier alone drains a 49 volume of ice above flotation equivalent to more than 3.5 m of sea-level rise (Rignot et al., 2019), similar to the 50 entire WAIS (Greenbaum et al., 2015).” However, the value of 3.5 m for Totten is from Greenbaum et al (Rignot et al., quote a value of 3.85 m). The Greenbaum paper focuses specifically on the Totten Glacier, so is inappropriate as a citation for the WAIS.

Corrected.

- Figure 1, the bathymetry is not very clear – consider using colour spectrum, as well as shaded relief. Consider a cross section showing shelf and subglacial topography, grounding lines, etc.

We redrawn Figure 1 by removing the multibeam maps, which are now shown in Figure 2 and 3 and in the Supplementary Figure S2. Multibeam maps are now provided as a colour spectrum.

- Figure 2. Line location symbology are different and hard to see.

We modified them according to the Reviewer comment.

- There are a number of spelling and grammatical issues that should be addressed.

We apologize for these annoying errors, which we are confident to have solved.

Reviewer #2 (Remarks to the Author):

Footprint of sustained poleward warm water flow within East Antarctic submarine canyons

The data this manuscript have a relative high quality taking into account the remote area and the difficult to obtained such type of data. These data were obtained with the appropriate techniques as ultra-high resolution seismic (Sub-bottom CHIRP). The main fortress of the manuscript is the potential significance of the results. In this way, authors point out to periodic intrusion of relatively warm deep water onto the continental shelf as a potential cause of melting of ice-caps in Antarctica. This introduce that fact the deep oceanic circulation of warm currents around the Antarctica as a threat to Antarctic ice shelves ice-caps and glaciers grounded below sea level.

The major ice-sheets around Antarctica (e.g. Weddell & Ross Sea Ice-Caps) acts as the main source of cold deep waters distributed around the world's oceans, playing an important role in "cooling" the Earth during the last geological history, Miocene to present. After my own experience of having carried out eight oceanographic expeditions to Antarctic, I support that role of deep currents in ice-cap melting as observed in the Weddell Sea and, even more, into their role in the distribution of Antarctic deep waters as forming the great system of oceanic's world circulation.

The intrusion of warm water onto the continental, as proposed in this manuscript by accelerating the melt of ice-caps might increase the cold deep water as the formation of Antarctic Bottom Waters (AABW) and their influence in the global ocean circulation. At geological scale, during deglacial Pleistocene times, the rapid retreat of the Antarctic ice-caps have been explained by increasing of temperatures in the southern oceans, and finally resulting rapid sea-level rise as consequence of the rapid retreat of massive ice-caps.

The only weakness that I observe in this manuscript is how to explain the water depth as much of 3,050 meters (4.2 s Two-way Travel Time) of the formation of the plastered drifts into deep incised S to N canyons as consequence of along-slope currents as the Circumpolar Deep Water (CDW). The plastered drift seems clearly associated to “upwelling currents” rising along a S to N submarine canyon and deflected to left by Coriolis effect. Authors explains these upward currents as results of semi-permanent cyclonic gyres formed by the interaction of cross-slope CDW current with the incision of the deep submarine canyons.

In my opinion, it seems much more difficult to explain the plastered drifts formed along the S-N canyons as results of orthogonal currents flowing in along-slope direction than from “upwelling” of N-S sourced deep waters.

It has been also proposed other mechanism to explain the spatial and temporal connection between upwelling of CDW and cascade of DSW (Dense Shelf Water) transport formed beneath ice-sheet. In this way, as a pulse of DSW descends the continental slope, the dense water displaces less dense water within the canyon allowing upwelling of CDW warm waters (see Morrison et al. 2020).

Probably, authors should improve the explanation(s) and discussion on the mechanism proposed dealing with the formation of cyclonic gyres enough or not to transport “upwelling” CDW warm waters onto the shelf.

Nevertheless, given the potential significance of the results, I strongly recommend the publication of this manuscript after making some further explanation and discussion on the proposed mechanism of intrusion/upwelling of warm waters onto the shelf.

We thank the Reviewer for all the above comments, which highlight that we were not clear enough in the discussion of each of the aspects he/she arose. As the Reviewer will notice, we have almost completely re-written the discussion, which, in our opinion, is now more clear in addressing their comments.

Reviewer #3 (Remarks to the Author):

Review for Nature Communications 463358, "Footprint of sustained poleward warm water flow within East Antarctic submarine canyons". Overall, I see no Major Weakness in the manuscript or research presented therein. First I'll summarized some overall thoughts before several minor weakness and line-by-line comments/edits.

Overall, this work represents an incremental but important step in Antarctic ice-ocean interactions, i.e. presenting new evidence for sustained meridional flow in two East Antarctic systems. The material is mostly presented in a clear fashion, and the research methods are sound - I do not have any criticisms of either their stratigraphic or hydrographical interpretations. In fact, I think they are exceptionally clear examples of the

hypothesis processes at play, and illustrates how to tie oceanography and solid-earth disciplines. However I have a few concerns about the overall impact of the work. As the authors themselves state, their work confirms that southward flow can be inferred using multichannel seismic/CHIRP, something that has been postulated by theory and confirmed elsewhere in Antarctica. While this is certainly a finding worthy to share with the community, for me it does not transform our thinking of the processes at play or the fate of glaciers in East Antarctica. The reader is told that the poleward flow of CDW to the ice sheet is critical for projecting future ice mass loss and sea level rise, but in its present form, the only result from this paper related to these topics are that the identified features "represent at least a ca. 440 to 1300 kyr long record of southward flowing currents". How does this help improve/understand projections of future glacial melt and SLR? Furthermore, although the CTD analysis is fairly sound, single hydrographic profiles cannot in any way 'confirm' a sediment record thousands of years long. In summary, this work, as submitted, provides very clear and compelling arguments for extended periods of southerly flow along these canyons, but I came away unconvinced of its importance to the stated motivations (i.e., ice melt and sea level rise)

In our opinion, in this revised version of the manuscript we better emphasized the main outcomes of our study, which we consider both novel and of broad interest for interdisciplinary studies of the glaciological, oceanographic, climate and geological communities:

1. Several areas of the Antarctic margin are still largely unexplored, and in some of them only one type of data, e.g., seismic data has been collected. This is the case for large parts of East Antarctica, where seismic data and "only" a few oceanographic measurements and sediment cores are available. In these areas, the occurrence of sedimentary features similar to those we unveiled here, could represent a key morpho-sedimentary proxy documenting sustained southward bottom flows and associated transport of warm CDW across the continental rise and slope and ultimately its intrusion onto the continental shelf.
2. The role of submarine canyons as effective conduits in funneling warm CDW towards the continental shelf. They are thus key regions for investigating the mechanisms governing CDW intrusions and the role it plays in Antarctic ice sheet (in)stability with implications for Global Mean Sea Level.

Ln 24: Subject?

We did not have understood what the Reviewer is asking here, sorry. Possibly, "marine-grounded glaciers" help this sentence to be clearer?

Ln 25: What is the justification of looking at these two glaciers?

Because of the limited amount of words allowable in the Abstract, we could not add any statement concerning this Reviewer comment. Instead, we added some paragraphs related to the importance of studying these two glaciers in the Introduction and in the Conclusions

Ln 26: What about the canyons isy being 'documented'?

We have modified this sentence

Ln 31: Does it say something about SLR directly, or just indirectly through glacial melt?

We added "consequent" before "sea level rise".

Ln 59-61: This sentence is ambiguous. As written it sounds like there are “Dalton Polynyas” in “other Antarctic regions”. Also, are you saying that the Dalton Polynya hasn’t formed? Hasn’t produced DSW?

We have slightly modified this sentence in order to make it clearer.

Ln 69: The bathymetry resolution in some areas seems too low. Can you use Bedmap3? It appears to be available via SCAR as of this summer.

Yes, we are aware about Bedmap 3. However, at the time of the manuscript submission, the Bedmap 3 gridded data were not available, and they are still to be provided, as reported in the related website. (<https://www.bas.ac.uk/project/bedmap/#data>)

Also, the ‘Depth’ scale doesn’t mean much, and topography looks ‘inverted’ in 1b inset, maybe due to hillshade angle?

According to both Reviewers 1 and 2, we substituted the previous images of the multibeam bathymetry with new ones, where we have used a colour spectrum.

Finally, perhaps this Figure 1 is not needed? All the reader gets (in its present form) is the location and trough/slope shape overview. Readers already get the bathymetry in more detail in Figs 2 and 3. So consider eliminating Fig 1 and add a circum-Antarctic Overview inset to each Figs 2 and 3?

We modified Figure 1 by removing the new multibeam maps, which are now shown in Figures 2 and 3 and Supplementary Figure S2.

Ln 74: Really ‘one of the largest’?

We have modified this part

Ln 93: “are”?

We are referring to the acoustic facies, which is singular, so that “is” would be correct.

Ln 104: “argue”?

Ok, modified.

Ln 105: How wide is the canyon/current? Is the Rossby radius of deformation relevant here? I know it is difficult to calculate given the dearth of measurements, but can it be estimated?

To reply to this request, we calculated the Rossby radius from the Brunt-Väisälä frequency profiles of the CTD stations collected during the austral summer season and described in detail in Bensi et al. (2022). The first baroclinic modes have Rossby radii between 4 and 5.5 km. Regarding, in particular, the two deep profiles within the canyons, 7 and 8, they have values of 5.4 and 5.6 km respectively, that scale down to smaller values for higher modes. The width of the canyons we consider is comparable or larger than these values, hence we may deduce that the canyon morphology may interact with both smaller dynamic features, including small scale eddies, internal waves, and with the flow subject to the Coriolis effect. We therefore, integrate the text accordingly at the end of the Section Results (“The vertical gradients.... the width of the canyons. “), and in the Section Discussion (“Comparing the first Rossby radius....observed deposits. “)

Ln 113: “Only”? Is this strong of a statement defensible?

We have almost completely rewritten the parts of the results and discussion dealing with the origin of the sediment drifts.

Ln 119: Capitalize? And add “.”

Done

Ln 131: In addition to the cartoon in Figure 5, the discussion of these drift and levee features could benefit from a plan view zoom-in of a single feature (like 10 km wide version o)

Done. We have added an annotated zoom of the multibeam map in Figure 5.

Ln 143: It would be VER helpful for a zoomed in location plot for the CTDs shown in Figure 4, perhaps as an inset?

Done

Ln 144: “Potential” temperature? (theta)

Corrected

Ln 146: Apostrophe?

We have slightly modified this part of the sentence

Ln 153: As written, the “properties” access the continental shelf. I suggest “modifying the properties of CDW accessing the continental shelf.”

Ok, modified accordingly.

Ln 163: Is this specific to “East” Antarctica?

We modified this part.

Ln 165: “barotropic”?

Corrected

Ln 173: Remove comma.

Done

Ln 184: Interesting! I was wondering about ‘timing’ earlier in the paper. Is there a way to elevate this point or move it up?

Since the age model is derived from previous studies on the dated sediment cores, and is compared to other study areas, we would prefer to leave information about “timing” here.

This leads me to ask, can you compare southward flow during the different epochs? Even if relatively? Was it stronger/weaker or more/less consistent?

Unfortunately, no sediment cores have been collected so far at or in close proximity of the sedimentary features we discussed here. We would expect variations in these southward flows in relation to e.g., the Southern Annular Mode, since winds associated with it favor upwelling of (warm) deep water onto the continental shelf causing oceanic upwelling of CDW along the Antarctic continental shelf. At the moment, in our knowledge, no analogues of our study cases do exist, thus highlighting the novelty of our contribution, and, at the same time, the need of further investigations, which we are confident to perform in the framework of the new, recently approved PNRA DIONE (Dynamic behavior of the East Antarctic Ice Sheet in the Sabrina Coast; PI: F. Donda) project.

Ln 196: What does “rugged mean here? Have you made the case why it’s relevant?

“Rugged” in the sense of “rough”. We added the reference of Uenzelmann-Neben (2006), who highlights the major role of the seafloor bathymetry in the formation of the Antarctic Peninsula sediment drifts.

Ln 199: I don’t get the point of this short paragraph. Either clarify, or eliminate.

We think this is an important aspect: several areas of the Antarctic margin are largely unexplored, and in some of them only one type of data has been collected. This is the case for large parts of East Antarctica, such as between 100° and 140° E, where seismic data and “only” a few oceanography measurements are available. The discovery of features such as those we discuss here could allow the hypothesis that there are southward-flowing currents, possibly transporting heat to the base of the glacier, as is the case with Totten Glacier. According to the Reviewer comment, we have changed this sentence slightly to clarify this concept.

Ln 212: This confuses me, as throughout the rest of the paper we are told that this southward flow is extremely persistent. If so, what is its role in “destabilization”?

We have slightly modified this sentence in order to make it clearer.

Ln 219: In general I really like this cartoon, however some things are not very clear, especially the “sediment drift” features.

We modified this figure, and, in our opinion, all the shown features are clearer now

Also, "glacials" to “glacial periods”?

We substituted “glacials” with “glacial periods”

Ln 222: Remove “were” and “giant”?

Done.

Ln 224: “flowing flows”? Redundant.

Ok, we have substituted “flows” with “currents”

Ln 237: This list is hard to follow and therefore has little meaning.

Ok, we have modified this part of the sentence

Ln 255: Add space.

This paragraph has been modified in several parts, so that the error has been corrected.

Ln 280: This seems like a less-detailed, redundant paragraph. Suggest removing it.

We think that this paragraph is important since we state that for both the study areas the same seismic velocities have been used for the time-depth conversion.

Ln 295: I think “easy” is a bit too colloquial (and subjective) for a research article.

We agree with the Reviewer comment, and we modified this sentence accordingly.

Ln 296: Too qualitative for me. Can you put numbers on this noise? If not, remove.

Ok, we removed this sentence

Ln 303: Define “DTM”.

Done

Ln 310: “going to be”? This is not good practice.

Data collected offshore the Ninnis Glacier will be made publicly available within four years of collection. We have slightly modified this sentence.

REVIEWERS' COMMENTS

Reviewer #1 (Remarks to the Author):

Review of revised manuscript "Footprint of sustained poleward warm water flow within East Antarctic submarine canyons"

The revised manuscript submitted by Donda et al., addresses most of the reviewers' comments, and presents a clearer argument for the presence of sediment drifts within the canyon systems of the Sabrina Coast region offshore Antarctica. The new CHIRP profiles show much better evidence for sediment drifts (albeit the figure resolution needs to be improved) than in the previous manuscript, which is critical to the central argument of the paper.

I still have a number of more minor comments that should be addressed prior to possible publication.

1. Better evidence is now presented for the presence of drift-like features, with both bathymetric and CHIRP profile data shown from both location. However, why are the drifts described in the CHIRP profiles shown only on small ellipses on the bathymetry map? This is highlighted in the figure below, using the uninterpreted MBES data from the supplementary materials. You can see clear bathymetric evidence for a larger drift system, but this is not interpreted the same way by the authors. This should be amended in the figures and the accompanying text

Small patch drifts interpreted above;
single elongate drift evident on bathymetry
on right

2. I still think that too much is made of the "paleo-configuration" vs present day. In particular, with reference to the canyon being narrower in the present day (e.g. lines 128-135). It seems to be clear that the canyon system does have large levees, which indicates that downslope sediment transport has been active at a certain point, and that there are also drifts forming to the east and within the canyons, which I accept are plausibly formed by up-canyon flow. But the inference that downslope processes were more common in the "paleo-configuration" (i.e. glacial period – because of increased dense water overflow?) than the present, or that the canyon was wider then, seems speculative without more evidence. It's quite possible that both processes are occurring simultaneously, rather than alternating on glacial-interglacial timescales as implied. It's ok to frame that hypothesis, as the data is not currently available, but they shouldn't be so conclusive here. Presumably this is one of the ideas that will be tested by their future cruise.

3. The authors provide more references to sedimentation rates elsewhere, but all the references are to levees/open slopes/sediment drifts, and they still do not clearly state that the sedimentation rates in the canyon could be very different (potentially by orders of magnitude). This uncertainty should at least be acknowledged in the relevant part of the discussion.

I also suggest that the authors publish the digital data described in this paper, and provide a link to the data in the paper, unless there is an embargo on releasing the data.

Reviewer #3 (Remarks to the Author):

Overall, much improved all around. I think the authors arguments are much more sound with the revisions. My only comments are "minor" and I recommend acceptance if they are addressed.

Overall, much improved all around. I think the authors arguments are much more sound with the revisions. My only comments are “minor” and I recommend acceptance if they are addressed.

1) Figure 2. Much Improved! However, I still have a few comments:

The “zoom in” on panels A and B help, but are not much more interpretable than the full views. Perhaps it is resolution in the submitted PDF, colors chosen, or maybe it needs to be zoomed in even more?

I remain confused by colors in Figure 2c. Perhaps this is up to the Art team at Nature, but in the present condition the colors used to delineate units does not match the legend/key. Text labels could go a long way here. Additionally, I personally think it is a poor choice to color “Westward-advected downslope flows” any color at all, as it looks like another unit seen in MCS data. Perhaps black-and-white hatch, or just have the northward arrow head and leave the flow blob out of the figure?

The profile locations on the map (2d) are repeat of labels used for subplots. For instance, there is a Profile C-D, but there is also a panel “C”. The same goes for profile A-B. This needs to be deconflicted. Also note that “E” and “W” mean East and West some places but the start and end of the profile in 2c elsewhere. All this to say this needs to be cleaned up to improve readability. Consider the standard “prime” method, e.g. A-A’.

2) A comment on this “Response to Reviewer”:

“Concerning the reference of Morisson et al., 2020, we realized that it does not fit as an analogue because the southward flows in the Ross Sea derive from the gravity currents due to dense water formation on the shelf, which does not occur in our study area. In the Ross Sea, the southward flow on the eastern side of the trough is a response to the northward flow associated with dense water overflow on the western side. Therefore it is a different system.”

Could “paleo-polynyas” on the Sabrina Coast have generated HSSW and associated gravity currents down slope, therefore invoking the Ross Sea as an analogue? This possibility is relevant to several aspect of this work. Do we know anything about this?

RESPONSE TO REVIEWER COMMENTS

Reviewer #1

The revised manuscript submitted by Donda et al., addresses most of the reviewers' comments, and presents a clearer argument for the presence of sediment drifts within the canyon systems of the Sabrina Coast region offshore Antarctica. The new CHIRP profiles show much better evidence for sediment drifts (albeit the figure resolution needs to be improved) than in the previous manuscript, which is critical to the central argument of the paper.

I still have a number of more minor comments that should be addressed prior to possible publication.

1. Better evidence is now presented for the presence of drift-like features, with both bathymetric and CHIRP profile data shown from both location. However, why are the drifts described in the CHIRP profiles shown only on small ellipses on the bathymetry map? This is highlighted in the figure below (separate document), using the uninterpreted MBES data from the supplementary materials. You can see clear bathymetric evidence for a larger drift system, but this is not interpreted the same way by the authors. This should be amended in the figures and the accompanying text

We thank the Reviewer for this comment, which highlights that we were not clear enough. The elongate feature, clearly visible on the MBES data represents the present-day eastern levee of the canyon. Sediment drifts occur further east, and due to their apparent width (2000-3500 m)-we do not have enough data to constrain the actual size- they are not clearly visible even on the enlarged, uninterpreted MBES data provided in the Supplementary. We traced the two ellipses in correspondence to the sediment drifts recognizable on the two CHIRP profiles crossing them.

We added a sentence on the caption of Figure 2 to make this aspect clearer.

2. I still think that too much is made of the "paleo-configuration" vs present day. In particular, with reference to the canyon being narrower in the present day (e.g. lines 128-135). This part of the discussion is crucial, especially to understand the relationship between the "paleo" and the "present day" configuration of the main canyon system, and between the present day, small canyon levee and the sediment drifts (please see also comment above). In the previous revision, we reduced the discussion on the paleo-configuration, but here each statement is fundamental to allow the Reader to clearly understand how the different sedimentary features are related to each other. For this reason we prefer to leave this part as it is.

It seems to be clear that the canyon system does have large levees, which indicates that downslope sediment transport has been active at a certain point, and that there are also drifts forming to the east and within the canyons, which I accept are plausibly formed by up-canyon flow. But the inference that downslope processes were more common in the "paleo-configuration" (i.e. glacial period – because of increased dense water overflow?) than the present, or that the canyon was wider then, seems speculative without more evidence.

The inference of downslope processes being more common in the "paleo-configuration" is a widely accepted model applied not only for the study area (Donda et al., 2020; O'Brien et al., 2020; Donda et al., 2023), but also for other East Antarctic continental margins (e.g. George V Land). Based on this model, the repeated advances of a polythermal ice sheet throughout the Cenozoic were able to deliver

high amounts of sediments at the shelf edge, from where they were transferred to the continental slope and rise via slumps and turbidity flows (see also Rows 123-125). In the seismic profile shown in Fig. 2c, the occurrence of a huge debris flow, already identified and discussed in Donda et al., 2008 (Palaeogeography, Palaeoclimatology, Palaeoecology), and interpreted as reflecting major gravity-driven events.

It's quite possible that both processes are occurring simultaneously, rather than alternating on glacial-interglacial timescales as implied. It's ok to frame that hypothesis, as the data is not currently available, but they shouldn't be so conclusive here. Presumably this is one of the ideas that will be tested by their future cruise.

Yes, as the Reviewer highlighted, we are confident that the sediment cores we plan to collect in the frame of the future cruise will help better constraining our hypothesis.

3. The authors provide more references to sedimentation rates elsewhere, but all the references are to levees/open slopes/sediment drifts, and they still do not clearly state that the sedimentation rates in the canyon could be very different (potentially by orders of magnitude). This uncertainty should at least be acknowledged in the relevant part of the discussion.

We have not found any study concerning sedimentation rates in canyons in either the Antarctica or the Arctic. For this reason, we used as reference the sedimentation rates in areas which are similar in terms of cored deposits (i.e., sediment drifts in Antarctica), water depth (sediment cores collected off George V land), and sedimentation rates of the two sediment cores collected in the study area. To acknowledge the uncertainty in the sediment rates, we took the minimum rates recorded in the above mentioned locations.

I also suggest that the authors publish the digital data described in this paper, and provide a link to the data in the paper, unless there is an embargo on releasing the data.

The link to the data collected off the Sabrina Coast (in 2017) is provided in the "Data availability" section, whereas the data collected off the Ninnis Glacier (two years ago) is under embargo until 2026. We slightly modified the data availability statement accordingly.

Reviewer #3

Overall, much improved all around. I think the authors arguments are much more sound with the revisions. My only comments are "minor" and I recommend acceptance if they are addressed.

1) Figure 2. Much Improved! However, I still have a few comments:

The "zoom in" on panels A and B help, but are not much more interpretable than the full views. Perhaps it is resolution in the submitted PDF, colors chosen, or maybe it needs to be zoomed in even more?

Yes, the resolution of figures inserted in the World file and then converted to pdf is much lower than the original one. In our opinion, it is important that this figure remains at it is, i.e., without any modification to the

size of each panel, for the following reasons: 1. in panels a and b, the larger scale portions of the CHIRP profiles help the reader to understand the overall configuration of the present day canyon system; 2. As mentioned in methods section, due to the great depth (most of the sediment drifts lie at ca. 3500-3650 m depth), the original seismic profiles have a large intertrace (20-22 m), which affect their resolution, partly improved by the applied processing. While the signal quality could not be excellent, the value of this dataset in recording such a small scale, very deep, key features is immeasurable.

I remain confused by colors in Figure 2c. Perhaps this is up to the Art team at Nature, but in the present condition the colors used to delineate units does not match the legend/key. Text labels could go a long way here.

Possibly, what get the Reviewer confused was the green texture used to underline the debris flow, which we have removed.

Additionally, I personally think it is a poor choice to color “Westward-advected downslope flows” any color at all, as it looks like another unit seen in MCS data. Perhaps black-and-white hatch, or just have the northward arrow head and leave the flow blob out of the figure?

We agree with the Reviewer comment, and we changed the color by using a grey patch and a grey arrow.

The profile locations on the map (2d) are repeat of labels used for subplots. For instance, there is a Profile C-D, but there is also a panel “C”. The same goes for profile A-B. This needs to be deconflicted. Also note that “E” and “W” mean East and West some places but the start and end of the profile in 2c elsewhere. All this to say this needs to be cleaned up to improve readability. Consider the standard “prime” method, e.g. A-A’.

We agree with the Reviewer comment, and we modified the labels of the seismic profiles with “A-A’”, “B-B’” and “C-C’”.

2) A comment on this “Response to Reviewer”:

“Concerning the reference of Morisson et al., 2020, we realized that it does not fit as an analogue because the southward flows in the Ross Sea derive from the gravity currents due to dense water formation on the shelf, which does not occur in our study area. In the Ross Sea, the southward flow on the eastern side of the trough is a response to the northward flow associated with dense water overflow on the western side. Therefore it is a different system.”

Could “paleo-polynyas” on the Sabrina Coast have generated HSSW and associated gravity currents down slope, therefore invoking the Ross Sea as an analogue? This possibility is relevant to several aspect of this work. Do we know anything about this?

As stated in the previous comments, the northward currents in the Ross Sea are due to gravity currents triggered by processes of dense water formation on the shelf, which is not the case in our study area, or at least to a lesser extent. This is therefore a different situation to that on the Sabrina coast in the present day. However, as suggested by the Reviewer, we cannot exclude that these conditions were different in the past, and that paleo-polynyas could produce dense water. Further investigations are necessary. Therefore, we are planning a new survey in this area to collect new sediment cores and oceanographic data that could help better answer this interesting question.